# Segmental Lung Recruitment in Patients with Bilateral COVID-19 Pneumonia Complicated by Acute Respiratory Distress Syndrome: A Case Report

**DOI:** 10.3390/medicina59010142

**Published:** 2023-01-11

**Authors:** Alen Protić, Matej Bura, Alan Šustić, Josip Brusić, Vlatka Sotošek

**Affiliations:** 1Department of Anesthesiology, Resuscitation, Emergency and Intensive Care Medicine, Faculty of Medicine, University of Rijeka, 51000 Rijeka, Croatia; 2Department of Clinical Medical Sciences II, Faculty of Health Sciences, University of Rijeka, 51000 Rijeka, Croatia

**Keywords:** acute respiratory distress syndrome, COVID-19 pneumonia, lung recruitment, intensive care, mechanical ventilation

## Abstract

Bilateral COVID-19 pneumonia is caused by severe acute respiratory syndrome coronavirus 2 (SARS-CoV-2) infection and usually leads to life-threatening acute respiratory distress syndrome (ARDS). Treatment of patients with ARDS is difficult and usually involves protective mechanical ventilation and various types of recruitment maneuvers. A segmental lung recruitment maneuver by independent lung ventilation has been described as a successful recruitment maneuver in patients with lobar pneumonia, and may, therefore, be useful for the treatment of patients with bilateral COVID-19 pneumonia complicated by ARDS in the critical phase of the disease when all other therapeutic options have been exhausted. The aim of this case series was to present a case report of four mechanically ventilated patients with severe bilateral COVID-19 pneumonia complicated by ARDS using the segmental lung recruitment maneuver. The effect of the segmental lung recruitment maneuver was assessed by the increase in PaO_2_/FiO_2_ ratio and the lung ultrasound (LUS) scoring system (0 points—presence of sliding lungs with A-lines or one or two isolated B-lines; 1 point-moderate loss of lung ventilation with three to five B lines; 2 points-severe loss of lung ventilation with more than five B lines (B pattern); and 3 points-lung consolidation) determined 12, 24, and 48 h after segmental lung recruitment. In three of four patients with bilateral COVID-19 pneumonia complicated by ARDS, an increase in the PaO_2_/FiO_2_ ratio and an improvement in the LUS scoring system were observed 48 h after segmental lung recruitment. In conclusion, the segmental lung recruitment maneuver in patients with bilateral COVID-19 complicated by ARDS is an effective method of lung recruitment and may be a useful treatment method.

## 1. Introduction

Severe COVID-19 pneumonia is a viral pneumonia caused by severe acute respiratory syndrome coronavirus 2 (SARS-CoV-2) or COVID-19, which was first reported in the Chinese city of Wuhan in 2019 [1]. SARS-CoV-2 belongs to the *Coronavirinae* family and has a large single-stranded RNA genome with 27 to 32 kilobases that encodes at least four structural proteins (spike protein S, membrane protein M, an envelope protein E, and nucleocapsid protein N) [2]. Spike protein S, located on the surface of SARS-CoV-2, is important for pathogenesis and infection. It mediates virus entry into cells throught the interaction with the ACE-2 receptor on the host cell. Once inside the cells, SARS-CoV-2 replicates and triggers a host immune response that causes various clinical manifestations, including severe pneumonia that can be complicated by life-threatening acute respiratory distress syndrome (ARDS) [3]. The management of patients with ARDS in the intensive care unit (ICU) is challenging and includes protective mechanical ventilation, prone positioning [4,5,6], and various types of recruitment maneuvers [7,8]. Many different recruitment maneuvers can be used to improve pulmonary function in patients with ARDS, but there are still no consistent data and results are inconsistent. Hodgson et al. [9] showed that a gradual increase in positive end-expiratory pressure (PEEP) during the recruitment maneuver is safe and can improve oxygenation in patients with ARDS, while another study showed increased mortality in a group of patients with ARDS treated with the recruitment maneuver [10]. A segmental lung recruitment maneuver with independent pulmonary ventilation was described in our previous case report as a successful recruitment maneuver in patients with lobar pneumonia [11]. In this case report, a segmental recruitment maneuver was used for four patients with severe bilateral COVID-19 pneumonia complicated by ARDS and atelectasis as secondary problems, which, unfortunately, are often responsible for a fatal outcome. The aim of this case series was to present a case study of four mechanically ventilated patients with severe COVID-19 ARDS complicated by atelectasis in the basal lobes of both lungs, who were treated by a segmental lung recruitment maneuver with independent lung ventilation, and to investigate the effect of this treatment on stabilizing rapid deterioration of respiratory failure.

## 2. Materials and Methods

### 2.1. Patients

We present a prospective series of four reverse transcriptase-polymerase chain reaction (RT-PCR) COVID-19 positive patients with severe bilateral pneumonia complicated by ARDS and basal lobe atelectasis of both lungs, diagnosed by computed tomography (Figure 1), in whom segmental lung recruitment using independent lung ventilation was performed as a last resort method to stabilize rapidly deteriorating respiratory clinical parameters [12]. All patients were treated at the COVID Respiratory Center of the Department of Anesthesiology, Intensive Medicine and Pain Therapy, Clinical Hospital Center Rijeka, Rijeka, Croatia, according to the guidelines for COVID-19 pneumonia and ARDS [13]. The study was approved by the Ethics Committee of the Clinical Hospital Center Rijeka, Rijeka, Croatia, in accordance with the World Medical Association Declaration of Helsinki “Ethical Principles for Medical Research Involving Human Subjects”. Informed consent was obtained from a family member for each patient. Patients were analgosedated by intravenous infusion of sufentanyl at a dose of 0.05–0.075 microg/kg/h (Alatamedics, Zagreb, Croatia) and midazolam at a dose of 0.04–0.2 mg/kg/h (Dormicum, Roche, Basel, Switzerland). General monitoring included control of arterial blood pressure, central venous pressure, and body temperature. All patients received thromboprophylaxis with enoxaparin sodium (Clexane, Sanofi-aventis Group, Paris, France) at a daily dose of 1 mg/kg and corticosteroid therapy with dexamethasone (Krka, Novo Mesto, Slovenia) at a dose of 8 mg/day. Age, gender, body mass index (BMI), height, body weight, presence of diabetes mellitus and arterial hypertension, and duration of mechanical ventilation before segmental recruitment of patients are listed in Table 1. Hemoglobin, hematocrit, potassium, sodium, and glucose levels, acid-base parameters, C-reactive protein (CRP), procalcitonin absolute, and leukocyte count were analyzed twice daily. In addition, daily ultrasound examination of the lungs was performed, using a SonoSite Edge II ultrasound, FUJIFILM SonoSite, Inc., Bothell, WA, USA, and the LUS scoring system recorded [14]. Bronchoscopy with sampling was performed to monitor the bacterial colonization of the airways and the possible development of superinfection. When the clinical condition and respiratory parameters worsened, patients were placed in the prone position and continuous muscle relaxation with rocuornim bormide (Organon, Kloosterstraat, The Netherlands) at a dose of 0.5 mg/kg/h was included in the therapy [10].

### 2.2. Methods 

All four patients experienced clinical deterioration of respiratory parameters, and enhanced segmental lung recruitment of the lower segments of both lungs, and the middle segment of the right lung, was performed, according to the previously described technique in [11]. In brief, patients were additionally analgosedated for the procedure by administering an intravenous infusion of sufentanyl up to 0.075 microg/kg/h (Alatamedics, Zagreb, Croatia), and they received the neuromuscular blocker rocuornim bormid (Organon, Kloosterstraat, The Netherlands) at a dose of 0.6 mg/kg. A toilet of the tracheobronchial tree was performed before the procedure. Since atelectasis was predominantly present in the lower lung segments, a single-lumen pulmonary arterial catheter (PA) (Swan-Ganz, Edwards Lifesciences, Irvine, CA, USA) was inserted into the right bronchus distal to the bronchial separation for the right upper lobe, using the bronchoscope and a loop passed through the working channel of the bronchoscope, and the balloon was inflated to bronchial occlusion. Subsequently, the PA catheter was connected to the second ventilator (Drager Evita 2, Drager, Luebeck, Germany) via appropriate connectors. All four patients were ventilated for the 30 min in CPAP mode with a continuous pressure of 30 cm H_2_O and 30% oxygen-enriched air (the option to automatically turn on the Evita 2 ventilator during apnea was turned off). After 30 min, the PA catheter to the lower left bronchus was removed using a bronchoscope and a snare through its working channel, and this procedure was repeated for the next 30 min with a pressure of 30 cm H_2_O. The right upper lobe of the right lung and the entire left lung were continuously ventilated with a ventilator (Drager Evita XL, Luebeck, Germany) during the first phase (for the first 30 min), and the entire right lung and the left upper lung were ventilated during the second phase (for the second 30 min). The ventilator settings were as follows: PEEP 12, tidal volume (TV) 380 to 420 mL, frequency 20 to 24/min, inspiratory-expiratory ratio (I:E) = 1:1.5. After completion of the procedure, the PA catheter was removed, the patient was placed in the prone position for the next 12 h, and the previous lung-protective ventilation was resumed. All four patients had predominant atelectasis of the lower and partial middle lobes of the right lung and the lower segment of the left lung, and an additional 30 min of the procedure was performed per side, followed by an additional 20 to 30 min after repositioning of the PA catheter. During the procedure, all patients were hemodynamically monitored, and the data were recorded and are shown in Table 2. 

Lung ultrasound (LUS) was examined by means of a portable ultrasound system and a 2- to 5-MHz convex transducer (Sonosite Edge II, Fujifil Sonosite, Bohtell, WA, USA), and the LUS scoring system was applied in the supine position before (0 h), and 12, 24, and 48 h after, segmental lung recruitment. Patients’ posterior lungs were scanned in the lateral decubitus position on both sides, consecutively. LUS included examination of 12 lung regions: the upper and lower parts of the anterior, lateral, and posterior aspects of the left and right chest. Each region was scored from 0 to 3 points and the final LUS score was the sum of the points in all 12 regions and ranged from 0 to 36. The scoring was as follows: 0 points—presence of sliding lungs with A-lines or one or two isolated B-lines; 1 point—moderate loss of lung aeration with three to five B-lines; 2 points—severe loss of lung aeration with more than five B-lines (B-pattern); and 3 points—lung consolidation as shown in Figure 2 [14,15]. After the recruitment maneuver the CT scans were recorded to evaluate the persistence of atelectatic lung, as shown in Figure 3. For each patient, PaO_2_/FiO_2_ ratio and PaCO_2_ values were recorded before (0 h), and 12, 14, and 48 h after, the segmental lung recruitment.

### 2.3. Statistical Analysis

Statistical analysis was done using descriptive statistics and the results are presented in the tables. 

## 3. Results

The PaO_2_/FiO_2_ ratio, PaCO_2_ values, and LUS scoring before (0 h), and 12, 24, and 48 h after, segmental lung recruitment for each patient, are shown in Table 3. In all patients, a significant increase in PaO_2_/FiO_2_ ratio was observed at 12, 24, and 48 h after segmental improvement, when compared to the initial values (0 h) for all patients. In patients 1 and 4, PaCO_2_ values decreased to acceptable levels, whereas they did not change significantly in patients 2 and 3. Ventilation parameters of normoventilated lungs did not exceed the limits of acceptable peak pressure. The LUS score improved in three patients (1, 2, and 3), whereas the LUS score did not change significantly for the fourth patient.

All four patients were invasively hemodynamically monitored during the procedure, and the values of mean arterial pressure (MAP) and pulse values were recorded and are shown in Table 2. The MAP and pulse did not change significantly at any time point for any of the four patients. Airway pressures before and after segmental recruitment maneuvers are shown in Table 4.

## 4. Discussion

The management of patients with acute respiratory distress syndrome is always challenging. It is particularly difficult in patients with SARS-CoV-2 pneumonia, which is more often complicated by ARDS [16]. It has been reported that the prone position is effective in patients having COVID-19 pneumonia complicated with ARDS by increasing lung recruitment, decreasing atelectrauma and improving ventilation–perfusion matching when performed in the early stage of the disease [4,5,6]. In the four cases presented, the clinical course of the disease was unfavorable, despite appropriate treatment according to current guidelines and local protocol. Mechanical ventilation with standard methods of recruitment and ventilation in the repeated prone position was unsuccessful, and the patients developed a severe form of ARDS [17]. Computed tomography (CT) and ultrasound along the ARDS showed atelectasis mainly in the lower lung segments. At this stage of the disease, segmental lung recruitment was performed, which has already been described in the literature as a minimally invasive innovative procedure that could prevent the use of more invasive procedures, such as extracorporeal membrane oxygenator (ECMO) or even thoracic surgery [11]. Segmental lung recruitment increased the PaO_2_/FiO_2_ ratio in patients, indicating the success of the procedure itself and suggesting an increase in the lung area through which gas diffusion occurred. The same result was confirmed by the reduction of plateau pressure in the airways and the reduced need for PEEP after performing the maneuver. In patient 1, the PaO_2_/FiO_2_ ratio did not meet the criteria for severe ARDS, but there was a problem with ventilation and CO_2_ retention (PaCO_2_ = 24.4 kPa). After segmental lung recruitment, the proportion of successfully ventilated lungs from the same patient increased, and CO_2_ elimination was adequate after the first 12 h (PaCO_2_ = 7.5 kPa). Moreover, a 25–35% improvement in LUS score was observed in these patients within 48 h after the procedures, although it should be noted that the LUS score [18] is limited by the fact that the number of B-lines is influenced by many factors and should be used with caution [19,20,21,22]. Therefore, new tools, such as automatic deep learning-based algorithms, can be used to estimate whether a lung is recutable [23]. It is very important to emphasize that segmental lung recruitment had no effect on hemodynamic deterioration in patients, proving that segmental lung recruitment is a sparing and minimally invasive method [24]. Classical methods of lung recruitment are limited to a short period of time, usually up to 40 s, and usually are not successful in opening lung atelectasis. In addition, during the classic recruitment maneuver, ventilation is completely stopped, and hemodynamic repercussions occur due to the sustained high airway pressure. This leads to an increase in intrathoracic pressure and a decrease in cardiac input, which, in turn, leads to a decrease in cardiac output [24]. In contrast, segmental lung recruitment did not result in the aforementioned side effects, despite a longer duration of the maneuver of up to 30 min. Moreover, during segmental lung recruitment, the part of the lung less affected by atelectasis was continuously ventilated. Finally, all four patients survived 7 days after the procedures, although respiratory deterioration was rapid immediately before the procedure, and the survival rate at 28 days was 25%. Regarding long-term follow-up of survivors, unfortunately all patients died, due to sepsis and multiorgan failure as a late complication of severe COVID-19. In this regard, we must point out that our ongoing clinical trial had better results in terms of long-term survival. This observation may give us a definitive answer to the question of the efficacy of segmental lung recruitment, and future clinical trials should include a larger number of patients.

## 5. Conclusions

Segmental lung recruitment can be an effective maneuver that improves respiratory parameters and ultrasound scoring within the first 48 h of its performance and has no repercussion on hemodynamics like classical recruitment in mechanically-ventilated patients with pneumonia or ARDS.

## Figures and Tables

**Figure 1 medicina-59-00142-f001:**
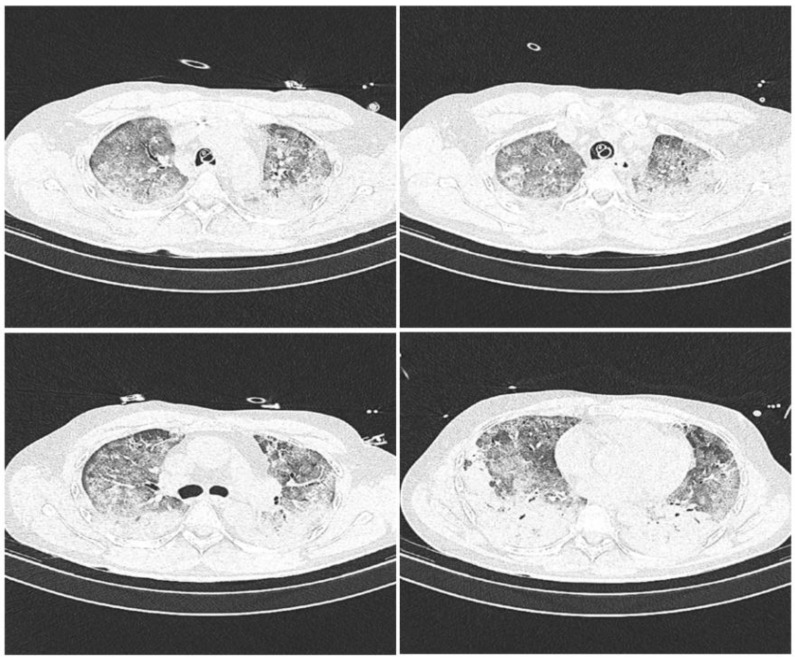
Computed tomography scans prior to segmental lung recruitment. CT scans showing different sections of lung: apical lung (**top left** picture), middle part of the lung (**top right** and **bottom left** picture) and basal lung (**bottom right** picture).

**Figure 2 medicina-59-00142-f002:**
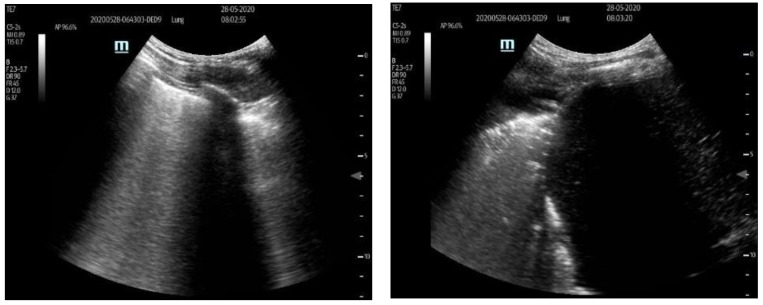
Ultrasound scoring system used to evaluate loss of aeration. On **left side** ultrasound score 2 showing severe loss of lung aeration with more than five B lines. On **right side** ultrasound score 3 showing lung consolidations.

**Figure 3 medicina-59-00142-f003:**
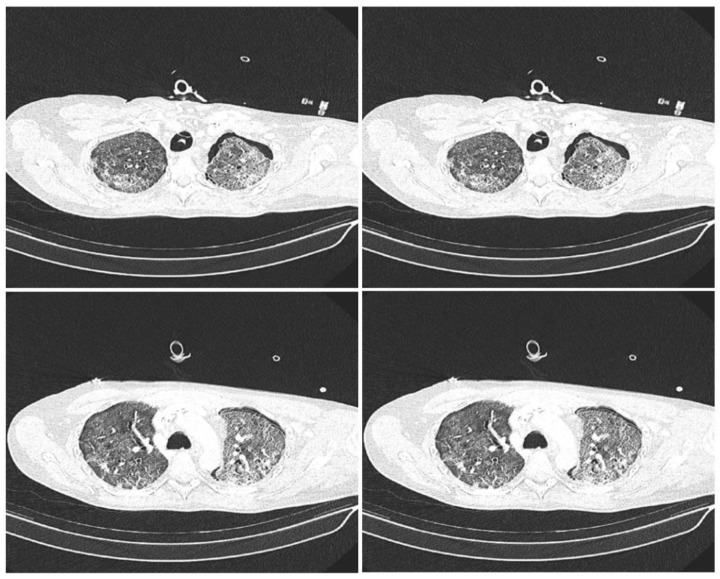
Lung computed tomography scans after the segmental lung recruitment CT scans showing different sections of lung: apical lung (**top left** picture), middle part of the lung (**top right** and **bottom left** picture) and basal lung (**bottom right** picture).

**Table 1 medicina-59-00142-t001:** Demographic data, co-morbidity and time spent on mechanical ventilation prior to segmental lung recruitment.

ID	Age	Gender	BW	BH	BMI	DM2	AH	MV
1	46	M	103	173	34.3	NO	YES	120
2	45	M	75	170	26	NO	NO	144
3	47	M	110	181	33.6	YES	NO	96
4	44	F	90	168	31.9	NO	YES	192

ID—patient identification number, BMI—Body mass index/30 kg/m^2^, BW—Body weight/kg, BH—Body height/cm, DM2—diabetes mellitus type 2, AH—arterial hypertension, MV—mechanical ventilation prior to recruitment maneuver in hours.

**Table 2 medicina-59-00142-t002:** Mean arterial pressure (MAP) and pulse values before (0 h) and 12, 24 and 48 h after performing segmental lung recruitment.

ID	MAP (mmHg)				PULSE (bpm)			
	0 h	12 h	24 h	48 h	0 h	12 h	24 h	48 h
1	77	108	100	71	113	116	118	93
2	82	104	115	105	69	74	77	70
3	82	109	70	87	88	110	99	80
4	92	86	78	83	100	100	70	70

ID—patient identification number, MAP—mean arterial pressure, bpm—beats per minute.

**Table 3 medicina-59-00142-t003:** The PaO_2_/FiO_2_ ratio, PaCO_2_ values and LUS score before (0 h), and 12, 24 and 48 h after, the segmental lung recruitment.

ID	PaO_2_/FiO_2_ (mmHg)				PaCO_2_ (kPa)				LUS			
	0 h	12 h	24 h	48 h	0 h	12 h	24 h	48 h	0 h	12 h	24 h	48 h
1	108	103	130	145	24.4	7.5	8.2	8.9	28	20	19	18
2	148	185	179	179	6.9	7.6	6.5	7.6	26	15	17	19
3	58	151	125	157	7.4	6.9	5.5	7.3	27	20	19	20
4	56	75	87	103	10.4	6.3	5.3	6.1	22	21	25	24

ID—patient identification number.

**Table 4 medicina-59-00142-t004:** Airway pressure values before (0 h), and 12, 24 and 48 h after, the segmental lung recruitment.

ID	PEEP (mmHg)				Pplat (mmHg)			
	0 h	12 h	24 h	48 h	0 h	12 h	24 h	48 h
1	16	12	12	12	28	24	23	23
2	14	14	14	14	24	24	25	24
3	15	14	14	14	30	30	29	29
4	14	12	12	12	28	26	26	25

ID—patient identification number, PEEP—positive end-expiratory pressure, Pplat—plateau pressure.

## Data Availability

The datasets used and analyzed in the present study are available from the corresponding author on reasonable request.

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
