# Peer review of "Segmental Lung Recruitment in Patients with Bilateral COVID-19 Pneumonia Complicated by Acute Respiratory Distress Syndrome: A Case Report"

_medicina, 2023, doi:10.3390/medicina59010142_

Round 1
Reviewer 1 Report (New Reviewer)
This case report is designed properly and written well, however, some of the points needs to be addressed before its publication. I have mentioned all the points below:1. There are few grammatical mistakes in the manuscript, hence I request authors to check the manuscript carefully before submitting the revised version of manuscript.
2. Please provide some information on the causative agent of COVID-19 in the introduction, I suggest reading and citing the article Aerosol transmission of SARS-CoV-2: The unresolved paradox. Trav Med Infect Dis. 37:101869
3. Methods are good opted for this research.
4. Results and discussion is ok.
5. Many of the references are not as per the format of the journal. Kindly update the same in the revisions.
Rest is ok.
Author Response
Please see the attachment.

Reviewer 2 Report (New Reviewer)
Please add:
How was the diagnosis of COVID-19 made?
Lung CT after segmental lung recruitment?
Outcome
Long-term follow-up of survivors
Please review and comment on the following references:
Fossali T.
Crit Care Med 2022; 50:723.
Langer T.
Crit Care 2021;25:128.
Ehermann S.
Lancet Respir med 2021; 9:1387.
Patel BV.
Intensive Care Med 2021; 47:549.
Maiello L.
Front Physiol 2022; 12:725865.
Author Response
Please see the attachment.

This manuscript is a resubmission of an earlier submission. The following is a list of the peer review reports and author responses from that submission.
Round 1
Reviewer 1 Report
This paper was to explore the effectiveness of segmental lung recruiting manoeuvre on four ARDS patients caused by COVID-19 infection, and it was concluded that it is an effective method of lung recruitment and can be useful method of treatment.There are many problems in this study. First, No typical ultrasound picture was provided, so it was impossible to determine whether the lung ultrasound was performed accurately or the results were reliable. Second, Because the expression of B-line is disturbed by too many influencing factors, the LUS score is questioned, and its accuracy is worth considering.Such as:
- Kameda T, et al Ultrasound B-line-like artifacts generated with simple experimental models provides clues to solve key issues in B-lines. Ultrasound in Med Biol, 2019; 45(7):16171626.
- Kameda T, et al.Simple experimental models for elucidating the mechanism underlying vertical artifacts in lung ultrasound:tool for revisiting B-lines. Ultrasound in Med Biol, 2021; 47(12) : 3543-3555.
- Matthias I, et al. Effect of Machine Settings on Ultrasound Assessment of B-lines. J Ultrasound Med, 2021;40:2039–2046.
- KamedaT, et al. The Mechanisms Underlying Vertical Artifacts in Lung Ultrasound and Their Proper Utilization for the Evaluation of Cardiogenic Pulmonary Edema. Diagnostics, 2022;12:252.
- Liu J. The Lung Ultrasound Score Cannot Accurately Evaluate the Severity of Neonatal Lung Disease. J Ultrasound Med, 2020; 39(5):1015-1020.
Reviewer 2 Report
Dear colleagues, dear Editor,
I appreciate the opportunity to review this manuscript.
The idea presented by the authors in this case series is interesting. However, the discussions regarding the concept of lung recruitment are highly controversial. If recruitment maneuvers are used and advocated in research papers, a comprehensive description of the physiological rationale, the methodology and the results is obligatory. Overstatements in discussion and conclusion must be omitted.
In line with these general statements, I have the following comments, most of which are major points.
Which definition of ARDS was used? Which imaging studies were used to confirm ARDS and regional lung deaeration, i.e. collapse or consolidation?
There is no Statistics paragraph, please add a description of the statistical methods used to support the statements made in your text.
Was this case study prospective or retrospective?
Please add relevant information on your patients, for example body weight, duration of ventilation before your recruitment procedure, tidal volume in ml/kg bodyweight, airway pressures, FIO2 (PaO2/FiO2 is not the same for 50 and 100% FiO2).
Please provide more data on mechanical ventilation. Did you really use CMV on the EVITA, i.e. IPPV? Together with the very low dose of sufentanil (please check if this is a mistake), I assume that spontaneous breathing efforts occurred during mechanical ventilation. In our institution, we experienced significant problems with patient-ventilator-asynchrony especially in young COVID-ARDS patients. Especially in early ARDS, spontaneous breathing (for example during APRV) can result in significant recruitment and it is difficult to understand the additional benefit of your invasive and experimental procedure in this situation. So please add a proper description of your patients, your physiological rationale and your setup.
A more comprehensive description of your respiratory and ventilatory management would really help understanding your procedure and the physiological rationale behind it. Do not get me wrong, I support recruitment maneuvers in recruitable lungs.
PEEP - positive end-expiratory pressure (not "peak expiratory pressure)
What is "double ventilation"? Wouldn't be "Independent lung recruitment" be a better description? “Independent lung ventilation” is a more commonly used English term, see article by S. Berg et al. World J Crit Care Med. 2019 Jul 31; 8(4): 49–58.
Did you really use a pulmonary artery (PA) catheter to achieve differential ventilation? The single lumen design described in your text is very unusual for PA-catheters, please provide manufacturer details of this catheter. Or did you use a bronchial blocker allowing gas flow to and from the isolated lung region? Ports of common PA catheters are very narrow and thus, together with the length of these catheters, flow through these ports (even for air/gas) is quite low and can be easily blocked. How did you ensure that the lower lung lobes actually experienced recruitment and received additional air/gas to fill the recruited lung units? How did you ensure that the inflow of fresh gas actually exceeded the resorption of oxygen from the "blocked" lower lobes?
Did patients receive neuromuscular blockers prior to the procedure? If not, how was spontaneous breathing activity excluded? If you actually used a small-bore PA catheter, a spontaneous inspiratory effort (especially the vigorous inspirations seen in young COVID ARDS patients) could create a large local transpulmonary pressure against a "quasi-blocked" airway. So, again, my question, how did you assess gas flow to the lung region blocked by the catheter?
The text on page 3/6, starting at line 127 is difficult to understand. Why atelectasis? Atelectasis is not the primary problem in COVID-ARDS? Why “additional 30 minutes”, what do you refer to? Did you repeat the procedure after turning your patients back into supine position? Please clarify!
Discussion and Conclusion: There are several statements made and conclusions drawn that are not supported (in any way) by your results. Be careful, your technique is NOT “non-invasive”, and your data derived from four patients do NOT allow any conclusions about safety or benefit.
Round 2
Reviewer 1 Report
This paper has a certain reference value, the method and results are reliable.However, it still needs to be revised before publication. (1)The abstract and preface need to be greatly simplified, and the introduction of COVID-19 can be weakened to focus on the purpose and significance of this paper.(2)There are only 4 cases in this article, so the title should be changed to Case report.
Reviewer 2 Report
The authors should not adequately respond to my comments. In fact, they chose to ignore the important questions or answered these points only very superficially. In short words, this time: There is still no definition for ARDS cited. It is still unclear whether you treated ARDS or atelectasis. Ultrasound has limited reliability in your setting. CT-images are not provided. You can't recruit a collapsed (atelectasis) or edematous (ARDS) lung (or lung region) with a CPAP of 30 applied through a very narrow catheter, which is not intended for this purpose. The pressure to recruit atelectasis is at least 40 cmH2O, see work of Dres. Rothen and Hedenstierna. The catheter you used is far too narrow, to allow enough gas inflow, the pressure drop has to be considered. It is expected, that authors try to discuss or respond to comments of reviewers. Just repeating the text that I commented on, is not enough. From the authors text, it remains unclear whether the patients in this case series or their relatives consented to this prospective? study? It remains also unclear, whether they suffered from edematous ARDS due to COVID, or had atelectasis along with a positive SARS-CoV2 test.
The manuscript still requires extensive revision.
Round 3
Reviewer 2 Report
The authors have not adequately responded to my comments. Instead of citing your previous case report, explain and discuss your method and the rationale behind it properly. I made enough suggestions. For example, why did you use an expensive, thin PA catheter instead of a bronchial blocker. The reasons for this off-label use should be explained along with my other points. There are many overstatements, still. Among others, the Compliance did not really change, which would be expected if recruitment had really happened. Improvements in gas exchange a minor, except for PaCO2 in Pat#1.
Author Response
We really appreciate your comments and we responded to all comments, questions, and ambiguities, but unfortunately, we noticed that our basic idea is not acceptable only for you as reviewer 2 while reviewer 1 did accepted the manuscript. Considering that our case series is an innovative approach to the unsolved problem of lung recruitment we did not find many studies to support our idea except our previous case report which is accepted and published. The statement that we wrote something irrational in our paper regardless of data we presented is not correct. We have accepted your suggestions and we are very grateful for those comments. Even these last suggestions that bronchial blocker could be a better tool than PA catheter is on the trial of our next study. But, here we have to correct you because the price of PA catheter is 100 euro in comparison whit bronchial blocker which cost 120 euro (we can prove the prices with an invoice). Finally, we are not sure that the price of devices used in clinical trials should cause negative criticism by reviewers. In this case series we tried to present four young people who were dying due to Covid 19 respiratory complication. According to our scoring systems all four patients had high prediction of death outcome within 24 hours, previously we started the procedure. We are aware that this is a small group of patients and that we need further investigation, but we think that our innovative procedure could represent the missing part of the puzzle in solving the lung recruitment problem. Therefore, we sincerely appreciate if you can revised one again the manuscript.